# Pellucidin A promotes antinociceptive activity by peripheral mechanisms inhibiting COX-2 and NOS: *In vivo* and *in silico* study

**Amanda Pâmela Santos Queiroz**[1◉], **Manolo Cleiton Costa Freitas**[2,3◉], **José Rogério A. Silva**[4], **Anderson Bentes Lima**[1,5], **Leila Sawada**[1], **Rayan Fidel Martins Monteiro**[1], **Ana Carolina Gomes Albuquerque de Freitas**[4], **Luís Antônio Loureiro Maués**[1], **Alberto Cardoso Arruda**[2], **Milton Nascimento Silva**[2,6], **Cristiane Socorro Ferraz Maia**[7], **Enéas Andrade Fontes-Júnior**[7], **José Luiz M. do Nascimento**[8], **Mara Silvia P. Arruda**[2], **Gilmara N. T. Bastos**[1]*

1 Laboratório de Neuroinflamação, Instituto de Ciências Biológicas, Universidade Federal do Pará, Belém, Pará, Brazil, 2 Laboratório Central de Extração, Instituto de Ciências Exatas e Naturais, Universidade Federal do Pará, Belém, Pará, Brazil, 3 Universidade Federal do Pará, Campus Universitário do Marajó-Breves, Breves, Pará, Brasil, 4 Laboratório de Planejamento e Desenvolvimento de Fármacos, Instituto de Ciências Exatas e Naturais, Universidade Federal do Pará, Belém, Pará, Brazil, 5 Laboratório de Morfofisiologia Aplicada à Saúde, Universidade do Estado do Pará, Belém, Pará, Brazil, 6 Laboratório Cromatografia Líquida, Instituto de Ciências Exatas e Naturais, Universidade Federal do Pará, Belém, Pará, Brazil, 7 Laboratório de Farmacologia da inflamação e comportamento, Faculdade de Farmácia, Universidade Federal do Pará, Belém, Pará, Brasil, 8 Laboratório de Neuroquímica Molecular e Celular, Instituto de Ciências Biológicas, Universidade Federal do Pará, Belém, Pará, Brasil

◉ These authors contributed equally to this work.
* bastosgnt@gmail.com

**Data Availability Statement:** All relevant data are within the paper.

## Abstract

*Peperomia pellucida* (PP) belongs to the *Peperomia genus*, which has a pantropic distribution. PP is used to treat a wide range of symptoms and diseases, such as pain, inflammation, and hypertension. Intriguingly, PP extract is used by different tropical countries for its anti-inflammatory and antinociceptive effects. In fact, these outcomes have been shown in animal models, though the exact bioactive products of PP that exert such results are yet to be discovered. To determine and elucidate the mechanism of action of one of these compounds, we evaluated the antinociceptive effect of the novel dimeric ArC2 compound, Pellucidin A by using *in vivo* and *in silico* models. Animals were then subjected to chemical, biphasic and thermal models of pain. Pellucidin A induced an antinociceptive effect against chemical-induced pain in mice, demonstrated by the decrease of the number of writhes, reaching a reduction of 43% and 65% in animals treated with 1 and 5 mg/kg of Pellucidin A, respectively. In the biphasic response (central and peripheral), animals treated with Pellucidin A showed a significant reduction of the licking time exclusively during the second phase (inflammatory phase). In the hot-plate test, Pellucidin A did not have any impact on the latency time of the treated animals. Moreover, *in vivo* and *in silico* results show that Pellucidin A's mechanism of action in the inflammatory pain occurs most likely through interaction with the nitric oxide (NO) pathway. Our results demonstrate that the antinociceptive activities of Pellucidin A operate under mechanism(s) of peripheral action, involving inflammatory mediators. This work provides insightful novel evidence of the biological properties of

**Funding:** The authors received funding for National Council of Scientific and Technological Development (CNPq), CAPES and Fundação Amazônia Paraense de Amparo à Pesquisa (FAPESPa).

**Competing interests:** The authors have declared that no competing interests exist.

Pellucidin A, and leads to a better understanding of its mechanism of action, pointing to potential pharmacological use.

## 1. Introduction

*Peperomia pellucida* (PP) is a plant used in folk medicine. It belongs to the Peperomia genus, which consists of approximately 1,600 tropical species with a pantropic distribution. It grows well in humid and shady conditions, presents succulent stems and heart-shaped leaves [1–3].

Traditionally, PP has been used in the treatment of a wide variety of diseases. Local communities of the pantropical have been using PP to treat several types of pain [4]. The areal parts of the plant have been described to treat various kidney diseases and decoction of the entire plant is used against poisonous animal bites by the Miskitu indigenous group of eastern Nicaragua [4, 5]. In Brazil, local communities have used PP in the treatment of abscesses, furuncles, inflammation in general, and hypertension [4].

In the ethnopharmacology field, there is growing evidence suggesting that PP has pharmacological properties. PP oils and extracts have been shown to inhibit both gram-positive and gram-negative microorganisms and human pathogenic fungi. The methanol extract exhibited a specific cytotoxic activity expressly against human cancer cell lines. PP has also presented potential analgesic activities [4].

Facing the fact that PP extracts and oils have been shown to present biological functions, the search for bioactive compounds in this plant is important, since this discovery will allow the development of new medicinal agents. In fact, previous studies with PP established the presence of several substances, such as tannins, flavonoids, among others [2, 6]. Interestingly, a chemical analysis of the methanolic extract from the aerial parts of PP has led to the isolation of a novel dimeric ArC2 compound, Pellucidin A [7]. Ayafor, suggest that these ArC2 dimers are products of a Diels-Alder reaction between two 2,4,5-trimethoxystyrene units, which literature also reports occurs in this plant [8]. However, Ayafor, names ArC2 dimers as bisnorlignans. The biological functions of Pellucidin A, have not been described so far to our knowledge [7, 8].

Intriguingly, PP is traditionally utilized in general cases of inflammation. Moreover, the use of PP in pain treatment led us to hypothesize that the PP compounds may interfere in the phenomenon of pain. These facts were supported by previous works, which showed that PP extracts present analgesic and anti-inflammatory properties in animal models of pain [1–10]. The exact substances in the PP extracts that are involved in these functions, as well as their mechanism of action, are yet to be discovered. Hence, in this work, we describe the biological functions of a PP compound, Pellucidin A, by evaluating its antinociceptive properties using *in vivo* models and clarifying the mechanism of action of this compound.

## 2. Materials and methods

### 2.1. Extract preparation

The specimen was collected from the Icoaraci district of Belém (Brazil). Experts from Emilio Goeldi Museum carried out the botanical identification. A voucher specimen was deposited under number 190136 IAN. The specimen's previously authorized collection was in November 2018, in Belém, from the Icoraci district, at geographic coordinates 1°28'09.9"S 48°29'40.5"W. The entire plant (Fig 1A) was washed thoroughly with water, and the leaves were then air-dried and ground into powder.

A

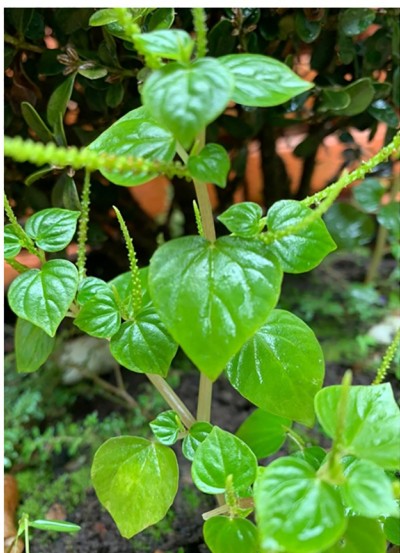

B

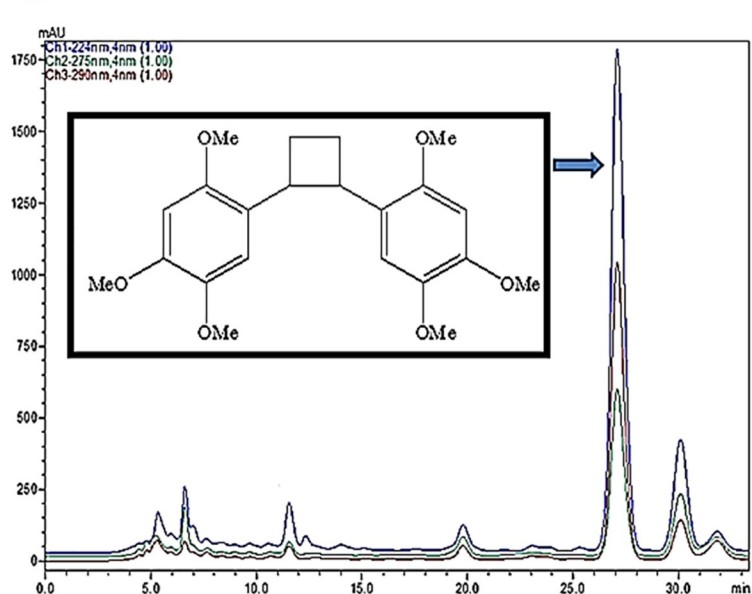

**Fig 1. Botanical image and Pellucidin A chemistry structure.** (A) *P. pellucida*, in preserved collection place, with inflorescence and leaves. (B) HPLC chromatograms of the extract of *P. pellucida* showing Pellucidin A peak.

The aerial parts of the dry and ground plant (450 g) were macerated at room temperature (25 ± 2˚C). The extractions were performed three times, for three days, with 96% ethanol (8 L). The hydroethanolic solution was concentrated in a rotatory evaporator under reduced pressure at 50–60˚C until it was completely dried, obtaining 32 g of crude extract. A sample of the ethanol extract (15 g) was subjected to percolation over a silica gel column using solvents with increasing polarity: hexane/EtOAc (9:1, 1 L, F1), hexane/EtOAc (7:3, 1 L, F2), hexane/EtOAc (1:1, 1 L, F3), EtOAc (1 L, F4), EtOAc/MeOH (1:1, 1 L, F5) and MeOH (2 L, F6).

## 2.2. Pellucidin A isolation

A 10 mg aliquot of the F5 fraction was analyzed by high-performance liquid chromatography (HPLC), solubilized in 1 ml $CH_3CN$, and then filtered through a membrane with a pore diameter of 0.25 μm. The chromatographic profile was analyzed using the following adopted method: Column Gemini $C_{18}$, 5 μm (250 x 4.6 mm) with a flow of 1 m/min; scanning with a wavelength between 200 and 400 nm (deuterium lamp), injection volume 20 μL, and isocratic $H_2O/CH_3CN$ (45:55) over 25 min. Based on the analytical methodology described, the F5 fraction was subjected to preparative HPLC, resulting in the isolation of Pellucidin A (57.3 mg).

## 2.3. Nuclear magnetic resonance analysis for Pellucidin A

The structural identification of Pellucidin A was carried out based on the analysis of spectral data from 1D and 2D NMR, and comparison with previously reported data [7]. The NMR spectra were recorded in $CDCl_3$ on a Varian MERCURY 300 MHz for $^1H$ NMR and $^1H$-$^1H$ COSY, and a Varian MERCURY 75 MHz for $^{13}C$ NMR and DEPT.

## 2.4. Animals

Experiments were conducted using adult male Swiss albino mice (8 to 12 weeks old; 20–35 g; n = 180), housed at 22± 2˚C under a 12/12 h light/dark cycle (lights on at 06:00), with free

access to food and water. Mice were obtained from the Evandro Chagas Institute (Belém, Brazil). All animal procedures described in this work were reviewed and approved by the animal ethics committee from the Federal University of Para, (CEUA-UFPA Nº 5671030216) and were carried out following the ethical guidelines for investigations of experimental pain in conscious animals.

## 2.5. Drugs, chemicals and reagents

The drugs and reagents used were morphine (Cristália, São Paulo, Brazil), indomethacin, NS-398 and L-NAME (Sigma Chemical Co., St. Louis, MO, USA), acetic acid (Vetec, São Paulo, Brazil), formaldehyde (Vetec, São Paulo, Brazil) and NaCl (Sigma Chemical Co., St. Louis, MO, USA). They were dissolved in saline solution, except for indomethacin which was dissolved in 5% NaHCO3 and Tween 80 plus 0.9% NaCl. Finally, the pellucidin A solution was prepared every day, before each experiment, with Tween 80 plus 0.9% NaCl. The final concentration of Tween 80 did not exceed 5% and did not cause any effect *per se*.

## 2.6. Open field

To assess the possible effects of the Pellucidin A on locomotor activity, mice were evaluated individually in an open field paradigm, as described [10], with some modifications. A wooden box (100 cm x 100 cm x 20 cm) with the floor divided into 25 squares was used. The mice were placed individually in the center arena of the open-field and behavioral parameters were counted manually for 5 min. Ambulation (number of squares crossed with four paws) and frequency of rearing (partial or total rising onto hind limbs) were measured. Male Swiss mice (n = 6/group) were treated intraperitoneally (i.p) with different doses of Pellucidin A (0.5; 1; or 5 mg/kg, i.p.), diazepam (2mg/kg i.p.), vehicle (saline solution 10 ml/kg) 30 min before the assay. Diazepam was used as a control to decrease locomotor activity [11].

## 2.7. Acetic acid-induced abdominal writhing

The test was performed as previously described by Sawada [12]. Nociception was induced with an intraperitoneal injection of 0.6% acetic acid (0.1 ml/10 g body weight). Male Swiss mice (n = 6/group) were treated intraperioneally (i.p) with different doses of Pellucidin A (0.5; 1; or 5 mg/kg, i.p.) one hour before acid acetic injection. To elucidate the Pellucidin A mechanism of action, cyclooxygenase (indomethacin 5 mg/kg; nonselective COX inhibitor), COX-2 (NS-398 10 mg/kg; selective COX-2 inhibitor) and nitric oxide synthase (L-NAME 5 mg/kg; nonselective NOS inhibitor) were used. All substances were administered intraperitoneally [12, 13].

A group of mice was treated with indomethacin (5 mg/kg) as a reference drug. Control animals received a similar volume of saline solution (10 ml/kg). Mice were individually observed in experimental acrylic cages. A hand-operated stopwatch was used to score the number of abdominal writhes. Writhing reflexes (characterized by the presence of abdominal muscles contractions) consist of inward outstretching of the hind limbs, hind paw reflexes, and extension of the whole body. The writhes were cumulatively counted over 30 min. A significant reduction in the number of writhes between the control and pretreated animals was considered indicative of antinociceptive activity.

## 2.8. Formalin test

The formalin test was carried out as previously described by Sawada et al, [12], with some modifications. The formalin solution is the algic agent. Licking is a rapid response to painful chemical stimuli that is a direct indicator of nociceptive threshold. The time that the animal

spent licking the injected paw, which is considered indicative of pain, was recorded for 30 min immediately following formalin injection. A significant reduction in the licking time was considered indicative of antinociceptive activity. Mice (n = 6/group) were treated intraperioneally (i.p) with different doses of Pellucidin A (0.5; 1; or 5 mg/kg, i.p.), and control animals received a similar volume of saline solution (10 ml/kg) 30 min before the formalin injection. Mice treated with indomethacin (10 mg/kg) or morphine (4 mg/kg, s.c.) were used as reference drugs. Morphine was administered subcutaneously (s.c.) to the morphine-treated group 15 min before the formalin injection.

### 2.9. Hot-plate test in mice

This test was adapted from Sawada et al, 2014 [12]. Animals were placed on a hot-plate set at 55± 0.5˚C. The hight temperature is the algic agent. The time between the placement of the mouse on the platform and shaking, licking of the paws or jumping was recorded as the hot-plate latency. Licking or jumping is a rapid response to painful thermal stimuli that is a direct indicator of nociceptive threshold. The animals were tested one day before the assay, and the mice with baseline latencies higher than 20s were eliminated from the study. Successive 30 min intervals before the administration of Pellucidin A (0.5; 1; or 5 mg/kg, i.p.) were applied. Saline solution was administered intraperitoneally. Morphine (10 mg/kg) was administered subcutaneously (s.c.) to the morphine-treated group. The reaction time was recorded when the animals licked their hind-paws or jumped.

### 2.10. Statistical analysis

Data in the text and figures are expressed as mean ± S.E.M. Statistical evaluations were performed using ANOVA, followed by Tukey test for individual pairwise comparisons, $p \leq 0.05$ was considered statistically significant.

### 2.11. Molecular docking

The crystal structures of inducible NOS (iNOS) (PDB code 1M8D [14]), endothelial NOS (eNOS) (PDB code 1M9J [14]), ancestral corticoid receptor (ACR) (PDB code 2Q1V [15]) and COX-2 (PDB code 4COX [16]) were obtained from the PDB website. Inducible and endothelial NOS enzymes contain chlorzoxazone (CLW) as a crystal inhibitor and heme group as a cofactor. The ancestral corticoid receptor has 17,21-dihydroxypregna-1,4-diene-3,11,20-trione (PDN) as a crystal inhibitor. For the COX-2 structure, indomethacin (IMN) is complex as an inhibitor. These inhibitors were used for a re-docking procedure to validate processes applied for molecular docking analysis. Then, the Pellucidin A was submitted for docking calculations.

The molecular docking studies were carried out by using the Molegro Virtual Docker (MVD) program [17]. This program has two docking search algorithms, MolDock Optimizer and MolDock SE (Simplex Evolution). The first is the default search algorithm in MVD [18], which is based on an evolutionary algorithm. However, MolDock SE performs better on some complexes where the standard MolDock algorithm fails [19]. Default parameters for the search algorithm were used to carry out molecular docking analysis. The detailed theory behind the MVD program and its characteristics are described elsewhere [17, 20].

## 3. Results

### 3.1. Isolation and identification of Pellucidin A

Isolation of Pellucidin A was possible using the HPLC method, where a single symmetric signal at the retention time of 27.09 min was obtained (Fig 1B). The absorbance of the eluting

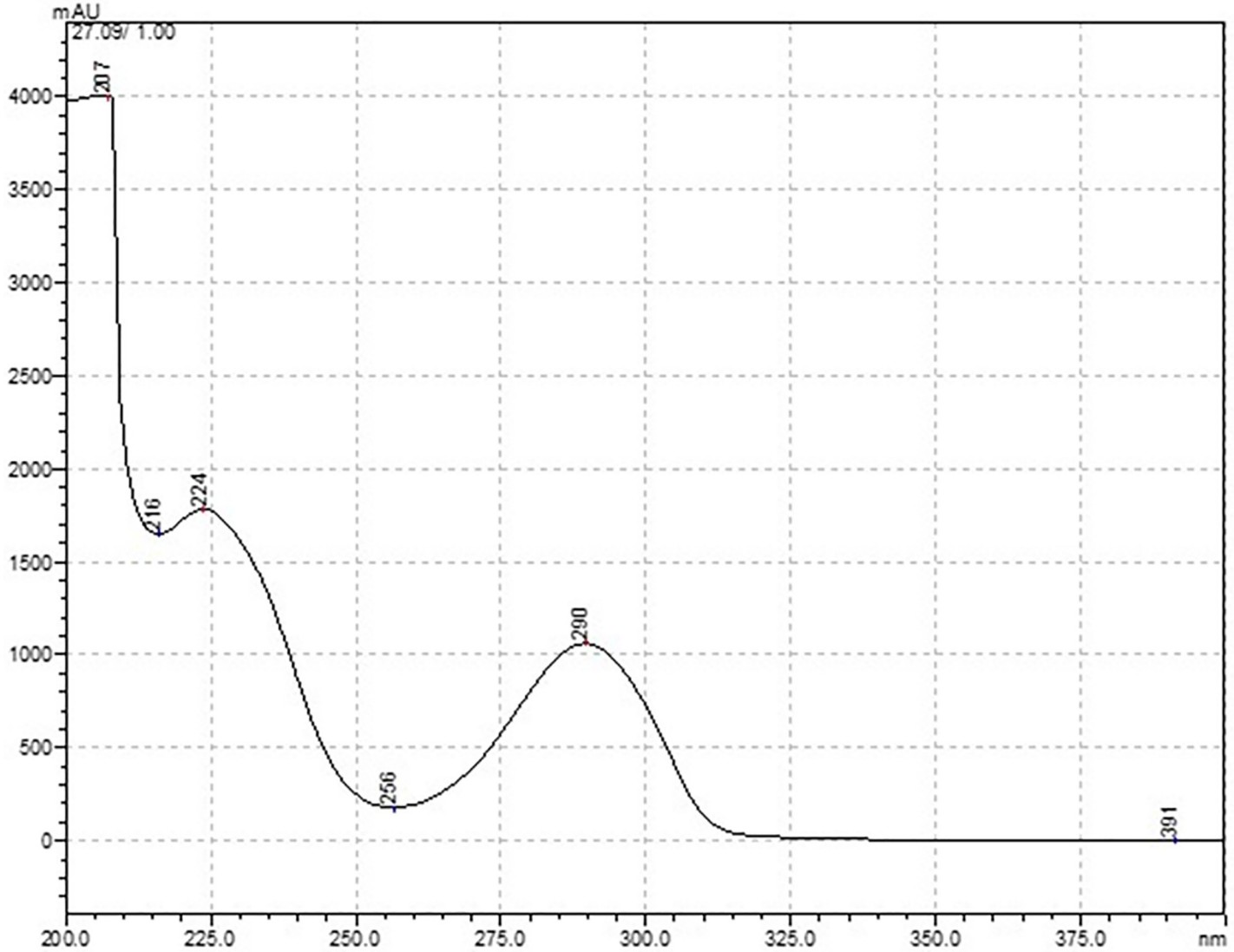

**Fig 2. UV-Vis plot spectra of the Pellucidin A.**

compound showed high intensity at 224 nm and relatively low at 290 nm (Fig 2). After isolation, NMR assay was performed to identify the substance, and the data obtained were compared with the literature data, further summarized in Table 1. These values match the chemical shifts reported in the literature [7].

## 3.2. Locomotor activity assessment

The effects of Pellucidin A on the exploratory and locomotor activities were measured in the open field assay (Fig 3). Compared with the control group, the number of crossings and the number of rearings were significantly decreased only in Diazepam-exposed mice. The animals that underwent treatment with Pellucidin A did not show a significant reduction in locomotor activity compared to the vehicle group. The Diazepam group was used as a reference drug since it induces alteration in the locomotor activity and on exploratory performance. Diazepan (2mg/kg) promoted sedation in the animals, decreased locomotion and exploratory behavior.

**Table 1. 1H and 13C NMR spectral data for Pellucidin A in CDCl3 at 300 MHz.**

| Position | Pellucidin A | | Literature[7] | |
|---|---|---|---|---|
| | $\delta_C$ | $\delta_H$ | $^*\delta_C$ | $^*\delta_H$ |
| 1/1' | 124.5 | | 124.9 | |
| 2/2' | 147.5 | | 147.8 | |
| 3/3' | 97.6 | 6.47 | 98.2 | 6.47 |
| 4/4' | 151.0 | | 151.3 | |
| 5/5' | 143.0 | | 143.3 | |
| 6/6' | 111.7 | 6.97 | 112.2 | 6.97 |
| 7/7' | 40.4 | | 40.6 | |
| 8/8' | 27.0 | | 27.0 | |
| OMe-4/4' | 56.3 | 3.74 | 56.3 | 3.74 |
| OMe-2/2' | 56.6 | 3.84 | 56.6 | 3.84 |
| OMe-5/5' | 56.1 | 3.85 | 56.1 | 3.85 |

## 3.3. The antinociceptive effects of Pellucidin A in acetic acid-induced constriction

Pellucidin A (0.5–5 mg/Kg, i.p.) produced inhibition of acetic acid-induced writhing response (Fig 4). Pellucidin A treated animals showed a reduction of 43% and 65% when treated with 1% and 5%, respectively. The number of writhings in animals treated with 0.5 mg/kg of Pellucidin A was not significantly different from the control group (p>0.05). Among the Pellucidin A treated groups, the group subjected to a 5mg/kg dose was the only group that did not show any statistical difference with the positive control indomethacin.

## 3.4. Elucidation of the antinociceptive effects of Pellucidin A in the formalin model

Pellucidin A antinociceptive effect could be interfering in two neuronal pain mechanisms: central and/or peripheral. To confirm and elucidate these mechanisms, we used the formalin test. An injection of formalin (2.5%) leads to a biphasic licking response to the injected paw by the animal. As shown in Fig 5A, Pellucidin A did not significantly reduce the time that the animal spent licking the formalin-injected paw during the first phase of this test. Morphine-treated

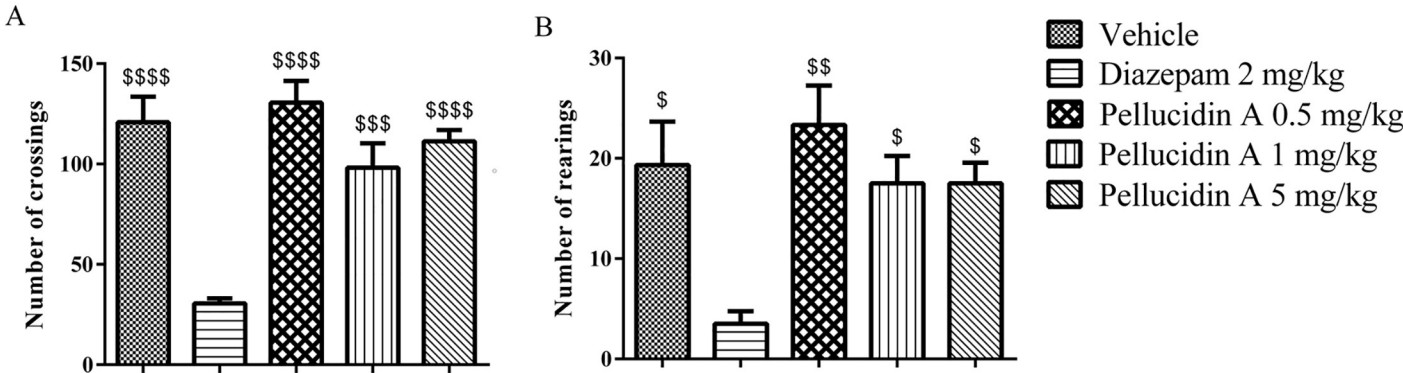

**Fig 3. Assessment of motor functions in controls and Pellucidin A-treated mice (0.5, 1 and 5 mg/kg).** (A) Spontaneous horizontal locomotor activity (A) and rearing activity (B) in the open field during the period of exposure. The Pellucidin A treated and Diazepam the positive control were used and compared with control animals for statistical analysis puporses. Data are means ± S.E.M. of n = 6 per group. The $ shown *p* value versus diazepam-injected. ANOVA with Tukey post hoc test was used.

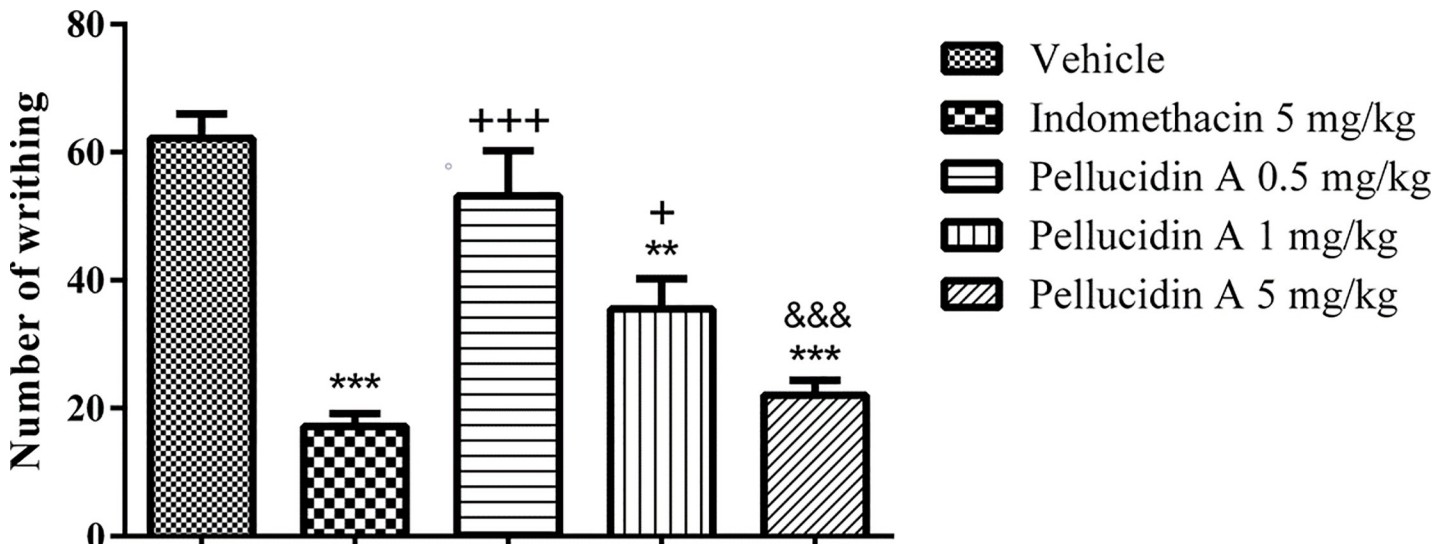

**Fig 4. Effect of Pellucidin A in acetic acid-induced nociception in mice.** Pellucidin A produced a dose-dependent inhibition of writhing response, that reached a maximum point at 5mg/kg. Data are means ± S.E.M. of n = 6 per group. The $ shown *p* value versus diazepam-injected. ANOVA with Tukey post hoc test was used. The * shown *p* value versus saline-injected, the + shown p values versus Indomethacin-injected, and & shown p values versus Pellucidin A 0,5 mg/kg.

animals manifested a significant reduction of the licking time during the first phase and indomethacin did not induce significant alterations in this parameter.

During the second phase (inflammatory) of this test (Fig 5B), Pellucidin A (5mg/kg) showed a significant antinociceptive effect (39.3± 4.0), reducing the licking time by 68%, when compared to control. Lower concentrations of Pellucidin A (0.5 and 1mg/kg) did not induce a significant reduction in the number of lickings when compared to control animals (Fig 5A). The positive control indomethacin (10 mg/kg) significantly reduced contractions (29.2 ± 3.9) when compared to control (123.5 ± 5.8), inducing contraction inhibition in nearly 76%. Among the Pellucidin A treated groups, the group dosed with 5mg/kg was the only group that did not show any statistical difference with the positive control indomethacin.

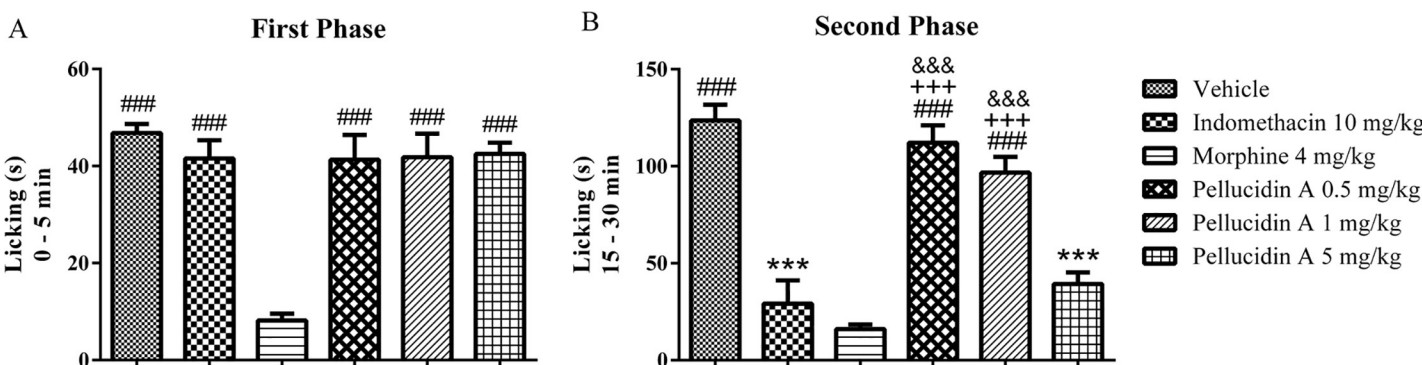

**Fig 5. Effect of Pellucidin A in formalin-induced nociception in mice.** (A)The activity of Pellucidin A in the first phase of a formalin induced-nociception test. (B) The activity of Pellucidin A in the second phase of a formalin induced-nociception. Data are means ± S.E.M. of n = 6 per group. ANOVA with Tukey post hoc test, was used. The * shown *p* value versus saline-injected, the + shown p values versus Indomethacin-injected, and # shown p values versus Morphine-injected, + shown p values versus Indomethacin-injected, and & shown p values versus Pellucidin A 5 mg/kg -injected.

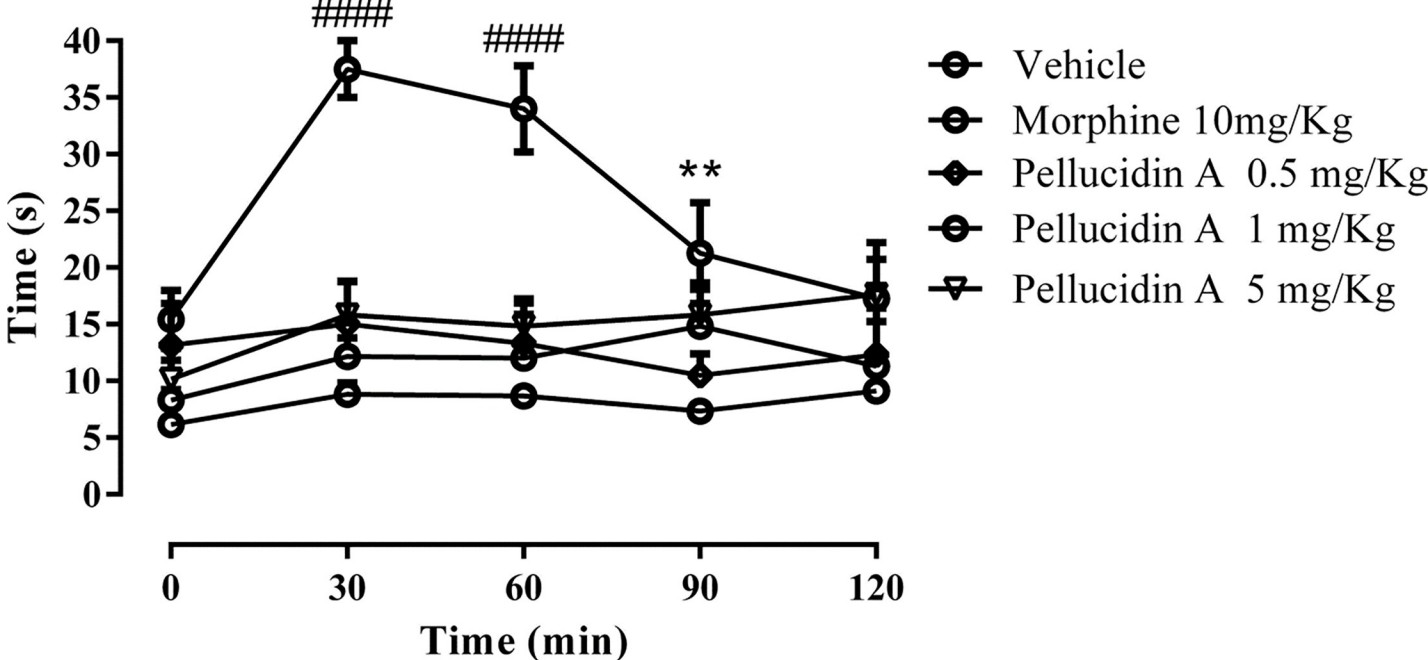

**Fig 6. Time-course of the effects of the Pellucidin A on thermal nociception.** Abscissa time (min) after Pellucidin A (i.p.) and morphine (s.c.) treatments. Ordinate latency time (s) for the response to thermal stimulation (55±0.5 °C, mean±s.e.m., $n = 6$) for each Pellucidin A dose. Data are means ± S.E.M. of n = 6 per group. ANOVA with Tukey post hoc test, was used. The * shown $p$ values from saline-injected.

### 3.5. Hot plate test in mice

As previous data showed that Pellucidin A may act on inflammatory and not on neurogenic pain, we decided to confirm this mechanism. We used the hot plate model to evaluate the supraspinal antinociceptive effects of the substance. According to the results in Fig 6, Pellucidin A did not induce a supraspinal antinociceptive effect in any of the different administrated concentrations (0.5, 1 and 5 mg/kg). On the other hand, morphine-treated mice (10 mg/kg) increased their latency time when compared to control.

### 3.6. Mechanism of action of the antinociceptive effects of Pellucidin A by combination with antagonists during acetic acid-induced writhing in mice

To evaluate which inflammatory mechanisms Pellucidin A was interfering with, mice were pre-treated with Indometacin, NS-398, and L-NAME (Fig 7). The pre-treatment with Indomethacin (27.5±2.9) and NS-398 (30.2±3.6) significantly reduced the number of contortions when compared to control (67.5±4.8), and their association with Pellucidin A did not induce a significant change in the number of writhings (22.5 ±3.3 and 20±5.4). On the other hand, the association of L-NAME with Pellucidin A induced a remarkable reduction in contortions (4.7 ±1.3) when compared to the L-NAME single treated group.

### 3.7. Molecular docking

The MVD program has been used successfully for molecular docking studies [21–24], a suitable procedure for validating this program through the analyzed systems is to carry out a re-docking by using bound crystal compounds. In this study, MVD results present in Table 2 describe good agreements between theoretical and experimental data, where the binding mode

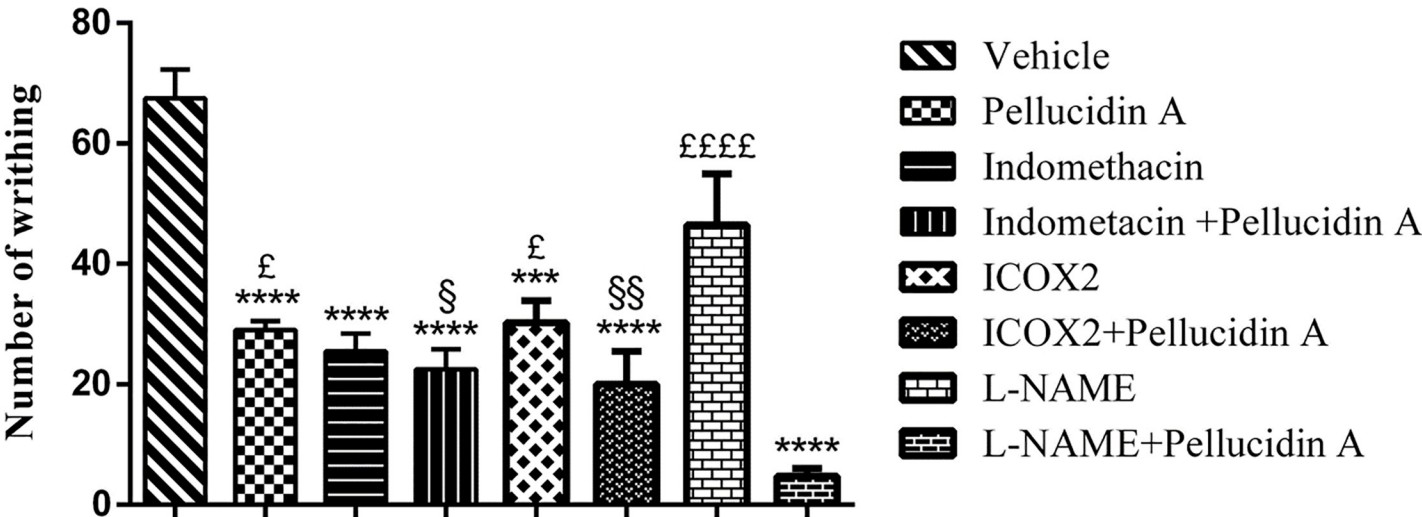

**Fig 7. Investigation of the mechanism of action of Pellucidin A-treated mice (5 mg/kg).** The effects of different inhibitors on the nociceptive activity of Pellucidin A in acetic acid-induced writhings. Animals that were pretreated exclusively with the inhibitors Indomethacin, NS-398 and L-NAME. The number of writhings was only slightly further reduced when Indomethacin and NS-398 were administrated with Pellucidin A, showing no statistic difference when compared to groups that were solely treated with inhibitor. On the other hand, Pellucidin A further reduced the number of writhings of L-NAME single treated animals, indicating that L-NAME and Pellucidin A presented a synergistic effect. Data are means ± S.E.M. of n = 6 per group. ANOVA with Tukey post hoc test, was used. The * shown *p* value versus saline-injected, the £ shown p values versus L-NAME+Pellucidin A -injected, and § shown p values versus L-NAME -injected.

for the ligands and their receptors are described successfully (Figs 8 and 9). Besides, atomic features occurring on the experimental complexes are elucidated by using molecular docking analysis.

## 4. Discussion

As *Peperomia pellucida* is a Neotropical plant largely used in traditional folk medicine [1], we were motivated to investigate whether its compound present any important pharmacological effect and elucidate its action mechanism. The central findings of this study revealed that anti-nociceptive activities of Pellucidin A operate under mechanism(s) of peripheral action through the inhibition of nitric oxide synthase and cyclooxygenase-2, in vivo and in silico, whithout central activity interference on motor function or exploratory coordination.

The findings revealed that Pellucidin A did not induce alteration in locomotor activity and exploratory performance. Therefore, the activity of Pellucidin A was evaluated in open field

**Table 2. Experimental and theoretical binding affinity values.**

| System | Experimental activity value | MOLDOCK scoring (kcal·mol$^{-1}$) |
| --- | --- | --- |
| eNOS-CLW | 8.70 μM | -120.10 |
| eNOS-PA* | - | -124.87 |
| iNOS-CLW | 14.10 μM | -114.20 |
| iNOS-PA | - | -125.21 |
| ACR-PDN | - | -120.70 |
| ACR-PA | - | -106.54 |
| uCOX-2-IMN | 0.96 μM | -155.86 |
| uCOX-2-PA* | - | -126.68 |
| cCOX-2-PA* | - | -107.187 |

*PA: Pellucidin A

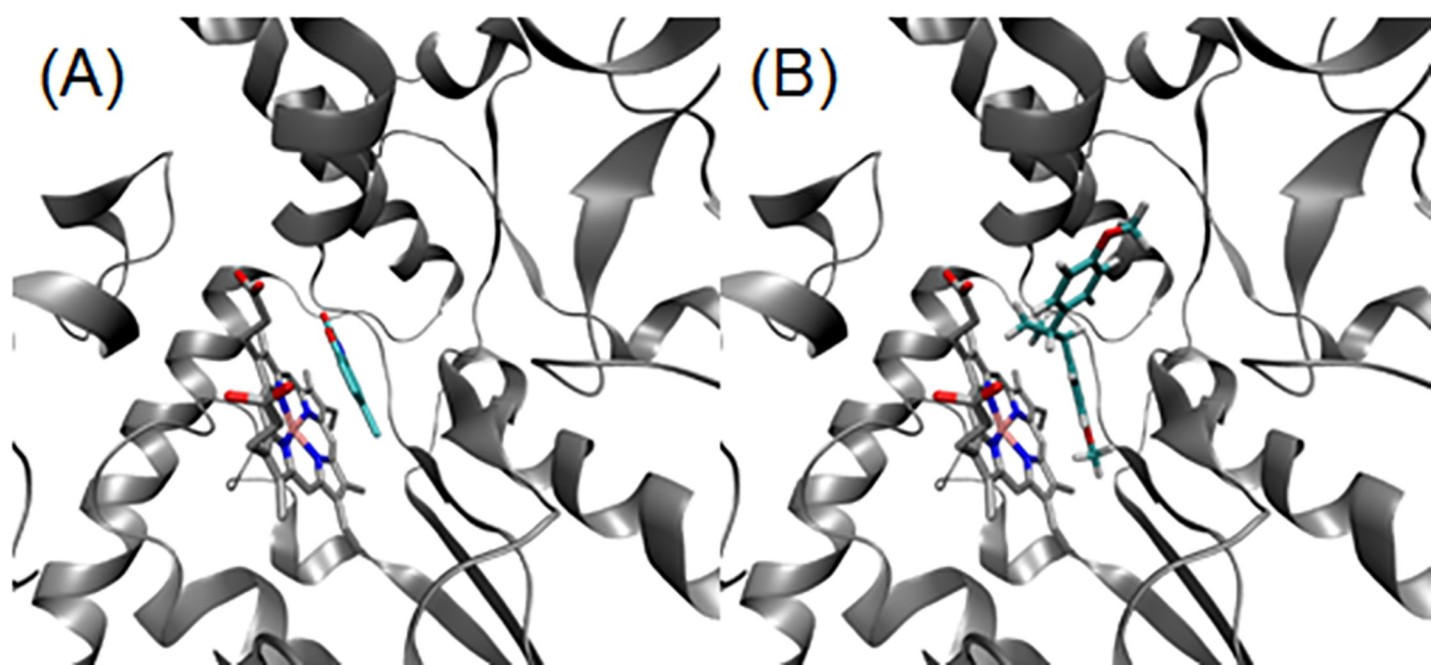

**Fig 8.** (A) eNOS-CLW and (B) eNOS-PA systems. Heme group on silver color and inhibitors on cyan color.

test that can be used as a model for screening central nervous system actions, providing information about psychomotor performance [10,11]. Pellucidin A isolated from *Peperomia pellucida* and administered intraperitoneally appeared not to have caused a central action because locomotor activity did not significantly decrease. Moreover, there was no interference in motor coordination or exploratory activity.Therefore, our results are consistent with previous findings that have previously demonstrated a mechanism of peripherical activity [1, 9, 25].

Because previous studies have shown that *PP* extracts present anti-inflammatory and analgesic properties [1, 9, 25], we investigated if Pellucidin A was involved in those properties. To do so, we performed the acetic acid-induced writhing response test, which is used to screen potential new agents, in particular, the analgesic and anti-inflammatory ones. This model induces the release of inflammatory-mediated substances that causes pain [26, 27].

Our data of acetic acid-induced writhing indicated that Pellucidin A reduced the number of writhings in a dose-dependent fashion, indicating that Pellucidin A presents potential anti-inflammatory and antinociceptive properties. In line with our results, De Fatima or Aziba, also observed a reduction in writhing utilizing *Pepperomia peluccida* extracts administrated with higher doses (70–500 mg/kg) [1, 9] however, these authors used different administration routes compared to our study. Interestingly, in our study, we found similar effects with purified Pellucidin A, one of differents compounds from *Pepperomia peluccida* extracts. This is the first study to investigate the pharmacological activity of purificated Pellucidin A.

To shed light into the potential central and peripheral effects of Pellucidin A, we conducted the formalin test. This nociception model allows differentiation between the central or peripheral analgesic effects. The first phase of this test reflects a central effect (neurological pain) and the second phase (inflammatory pain) implies a peripheral action involving, respectively, the direct stimulation of nociceptors and the involvement of inflammatory mediators, such as prostaglandin and NO [28]. As Pellucidin A significantly reduced the frequency of paw licking exclusively during the second phase of the test, it potentially exerts its antinociception

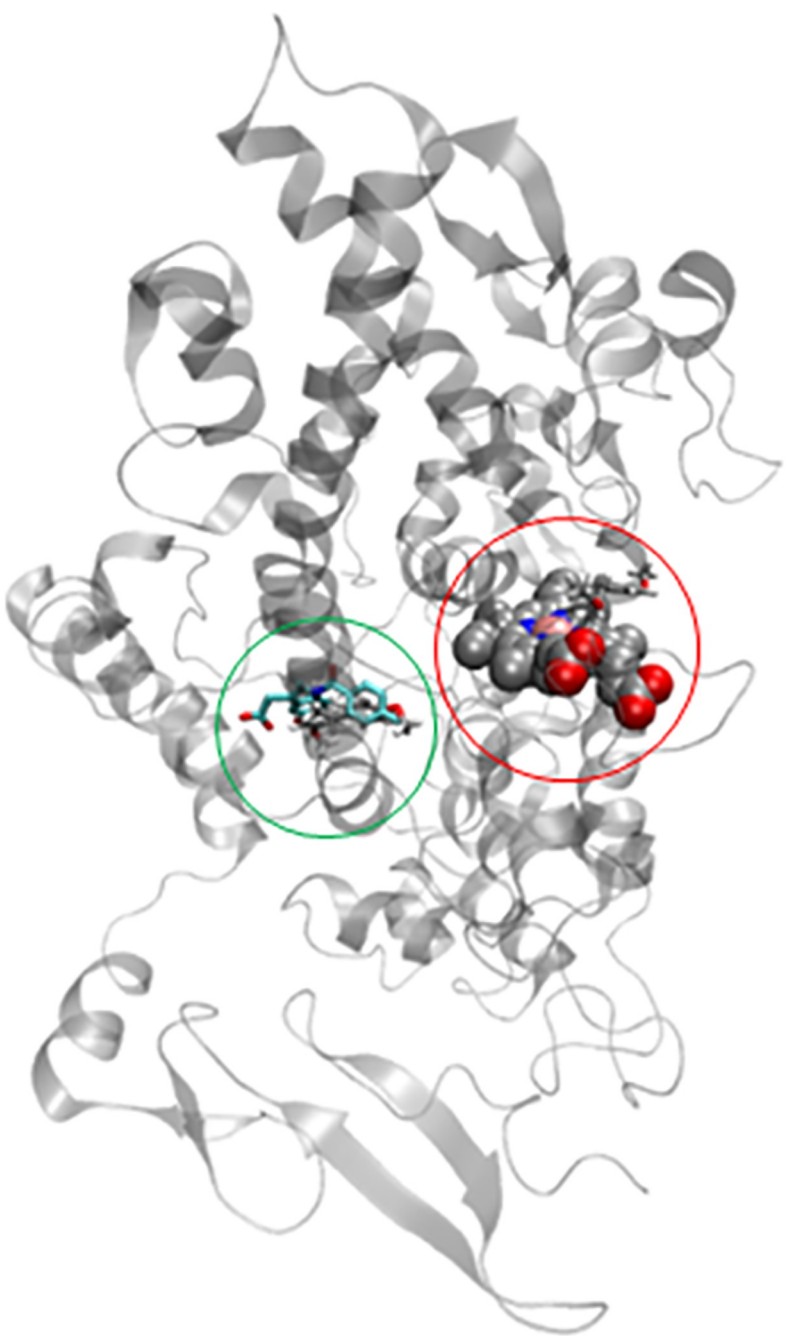

**Fig 9. Uncompetitive (green color) and competitive (red color) sites on COX-2 system.** PA inhibitor is present on both sites (silver color for C atoms).

properties through regulation of the inflammatory pathways. These results were further confirmed by the use of the hot-plate test, a central antinociceptive test. In this model, mediated primarily by supraspinal mechanisms, there is no involvement of inflammatory mediators [29]. Pellucidin A did not alter the latency time in hot-plate test; however, morphine did. These results suggest that the antinociceptive action of Pellucidin A occurs via peripheral pathways, instead of a central-acting mechanism.

Inflammatory pain is a complex process that can be triggered by several mechanisms. Particularly, prostaglandins are important mediators in inflammatory pain, partly by increasing the nociceptor sensitization [30]. Our results, evaluated in formalin and writing test, indicate anti-inflammatory and antinociceptive effects of Pellucidin A. According to De Fatima et al. [1], the aqueous extract of the aerial parts of PP exerts its anti-inflammatory and analgesic effects through the regulation of prostaglandins. Moreover, in carrageenan-induced paw edema, the extract exhibited anti-inflammatory properties from earlier time-points and throughout the phases of inflammation, suggesting that the *PP* compounds may act on different mediators of this process [1]. Our study indicate that Pellucidin A is the compound that promotes the above mentioned pharmacological effects. In addition, It is known that free radicals are involved in inflammation. Wei et al (2011) have shown that PP extract presents antioxidant properties, pointing that PP components may present anti-inflammatory, antioxidant and analgesic activities [31]. Our data suggest the Pellucidin A may present antinociceptive and anti-inflammatory activity but the antioxidant effect was not evaluated.

In our study, we revealed that Pellucidin A reduces the number of writhings in acetic acid and formalin induced pain models, possibly through interaction with Nitric oxide synthase. Consistent with this idea, the treatment with Pellucidin A showed a synergetic effect with L-NAME treatment, inhibiting contractions by 96%. Mice treated only with L-NAME presented 48% of contraction inhibition. This result follows our molecular docking model, which showed that Pellucidin A could bind to NOS, showing less energy demand to bind to iNOS.

Due to anti-iflammatory activity and mechanism experiment, we have used molecular docking methods in our work to better elucidate the Pellucidin A mechanism of action. These methods have been used successfully to propose the bind mode and interactions occurring between protein-inhibitor systems [20]. In this sense, we have carried similar analyses using Pellucidin A from chemical synthesis. To validate the molecular docking procedure as well as the theory applied by the MVD program, a re-docking calculation was computed using the crystal inhibitor found in each 3D crystal protein selected as a reference. Thus, comparing computational and crystal complexes revealed an excellent agreement between both models. In this way, the molecular docking procedure and parameters applied were used to compute the favored conformation of Pellucidin A in the binding site of selected proteins.

According to the re-docking results, for each inhibitor, the conformation obtained by molecular docking was in good agreement with its crystal conformation. Furthermore, Pellucidin A was also submitted to the same molecular docking steps. Our results show that it is complexed similar to the same binding pocket of the crystal inhibitors. Finally, our results suggest that the MVD program can reproduce suitable conformations for molecules like Pellucidin A compounds in complex with selected enzymes.

As for the COX-2 system, two potential binding pockets can be found for Pellucidin A as a potential inhibitor. The first is the same binding pocket of indomethacin crystal inhibitor (uncompetitive site, uCOX-2). The second is the binding pocket of the heme group (competitive site, cCOX-2). Both regions are large enough to accommodate Pellucidin A as a ligand. According to our results, Pellucidin A interacts better on the uncompetitive site.

The binding affinity values obtained through the MOLDOCK scoring function showed a good affinity of Pellucidin A as a potential inhibitor in all considered systems. By comparing Pellucidin A with crystal inhibitors on each system, it could act with more potency on NOS enzymes (eNOS and iNOS), on similar affinities. Particularly, for the COX-2 system, Pellucidin A shows strong interaction on uncompetitive binding sites, on the same tendency of IMN crystal inhibitor. This is very interesting since eNOS, iNOS, and COX-2 present the heme group as a cofactor. On the NOS systems, Pellucidin A interacts satisfactorily on the heme-

binding pocket, however, on the COX-2, the uncompetitive binding pocket is favorite for this molecule.

LaBuda *et al.*[32] showed that iNOS is involved in the acetic acid test in mice since the systemic administration of an iNOS inhibitor reduced the number of writhings. However, in a rat model, iNOS and nNOS inhibition does not seem to affect the number of writhings [33]. Nevertheless, in our mice and in silico model, we showed that Pellucidin A has the potential to act through interaction with NOS, indicating that iNOS plays a potential role in Pellucidin A antinociception function. iNOS is a molecule that is usually expressed after a stimulus. It is known that macrophages produce iNOS when stimulated with cytokines and/or other agents, and it is now accepted that iNOS can be produced by any cell that has been given the adequate stimuli [34].

The exact mechanism of action of Pellucidin A remains to be elucidated. De Fatima *et al.* [1] They reported that the PP extract did not block the arachidonic-acid induced edema formation, implying that the 5-lipoxigenase pathway is not involved with this extract's mechanism of action. Also, Nwokocha *et al.* [35] suggested a dose-dependent hypotensive effect of PP extract via nitric oxide-dependent mechanisms. If Pellucidin A is inducing these effects in those experiments, we might infer that Pellucidin A could be acting on prostaglandin synthesis and interacting with NOS. In fact, our study demonstrated a reduction of writhings in inflammatory pain models through potential interaction with iNOS.

We have demonstrated for the first time that Pellucidin A, a novel component from the widely tropically distributed plant PP, presents antinociceptive and anti-inflammatory properties, possibly inhibiting NOS on *in vivo* and *in silico* approaches. The interaction with COX-2 is less likely to occur when compared to the energy demanded the interaction between iNOS and Pellucidin A.

Therefore, Pellucidin A most likely promoted the antinociceptive activity by peripheral mechanisms, possibly through NO pathway, which supports PP ethnopharmacological use. In fact, previous work showed that PP has antioxidant properties [31]. Further studies must be performed to better characterize the exact mechanism through which Pellucidin A exerts its antinociceptive functions inhibiting eNOS and COX-2. The results showed in this work provide evidence that Pellucidin A, a novel component from the PP plant, may be applied as a potential new drug through inhibition of COX-2 and NOS.

## 5. Conclusion

In conclusion, we have demonstrated that Pellucidin A, a component from PP, exhibits dose-dependent antinociception when assessed in chemical but not thermal models of nociception in mice. Pellucidin A most likely promoted the antinociceptive activity by peripheral mechanisms and interaction with COX and NO pathways, which supports its ethnopharmacological use.

## Acknowledgments

Thanks the National Council for Scientific and Technological Development for their financial support and the South African Centre for High-Performance Computing (https://www.chpc.ac.za/) and University of Florida Research Computing (http://researchcomputing.ufl.edu) for providing computational resources.

## Author Contributions

**Conceptualization:** José Rogério A. Silva, Alberto Cardoso Arruda, Gilmara N. T. Bastos.

**Data curation:** Manolo Cleiton Costa Freitas, José Rogério A. Silva, Anderson Bentes Lima, Rayan Fidel Martins Monteiro, Ana Carolina Gomes Albuquerque de Freitas, Luís Antônio Loureiro Maués.

**Formal analysis:** Amanda Pâmela Santos Queiroz, Manolo Cleiton Costa Freitas, José Rogério A. Silva, Anderson Bentes Lima, Rayan Fidel Martins Monteiro, Luís Antônio Loureiro Maués.

**Funding acquisition:** Alberto Cardoso Arruda, Milton Nascimento Silva, Cristiane Socorro Ferraz Maia, Enéas Andrade Fontes-Júnior, José Luiz M. do Nascimento, Mara Silvia P. Arruda.

**Investigation:** Amanda Pâmela Santos Queiroz, Manolo Cleiton Costa Freitas, José Rogério A. Silva, Anderson Bentes Lima, Leila Sawada, Luís Antônio Loureiro Maués, Alberto Cardoso Arruda, Milton Nascimento Silva, Cristiane Socorro Ferraz Maia, Enéas Andrade Fontes-Júnior, Mara Silvia P. Arruda.

**Methodology:** Amanda Pâmela Santos Queiroz, Manolo Cleiton Costa Freitas, José Rogério A. Silva, Anderson Bentes Lima, Leila Sawada, Ana Carolina Gomes Albuquerque de Freitas, Luís Antônio Loureiro Maués, Milton Nascimento Silva, Cristiane Socorro Ferraz Maia, Enéas Andrade Fontes-Júnior, Mara Silvia P. Arruda.

**Project administration:** José Luiz M. do Nascimento, Mara Silvia P. Arruda, Gilmara N. T. Bastos.

**Resources:** Alberto Cardoso Arruda.

**Software:** José Rogério A. Silva, Anderson Bentes Lima.

**Supervision:** José Rogério A. Silva, Alberto Cardoso Arruda, Milton Nascimento Silva, Cristiane Socorro Ferraz Maia, Enéas Andrade Fontes-Júnior, Mara Silvia P. Arruda, Gilmara N. T. Bastos.

**Validation:** Manolo Cleiton Costa Freitas, Alberto Cardoso Arruda.

**Visualization:** Alberto Cardoso Arruda.

**Writing – original draft:** Manolo Cleiton Costa Freitas, José Rogério A. Silva, Leila Sawada, Gilmara N. T. Bastos.

**Writing – review & editing:** Rayan Fidel Martins Monteiro, Ana Carolina Gomes Albuquerque de Freitas, Gilmara N. T. Bastos.

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
