## [Decision Letter · Decision Letter 0]

2 Apr 2020

PONE-D-20-04760

Pellucidin A promotes antinociceptive activity by peripheral mechanisms inhibiting COX-2 and NOS:  In Vivo, and In Silico Study.

PLOS ONE

Dear Dr Bastos,

Thank you for submitting your manuscript to PLOS ONE. After careful consideration, we feel that it has merit but does not fully meet PLOS ONE’s publication criteria as it currently stands. Therefore, we invite you to submit a revised version of the manuscript that addresses the points raised during the review process.

We would appreciate receiving your revised manuscript by May 09 2020 11:59PM. To enhance the reproducibility of your results, we recommend that if applicable you deposit your laboratory protocols in protocols.io, where a protocol can be assigned its own identifier (DOI) such that it can be cited independently in the future. For instructions see: http://journals.plos.org/plosone/s/submission-guidelines#loc-laboratory-protocols

We look forward to receiving your revised manuscript.

Kind regards,

John M. Streicher, Ph.D.

Academic Editor

PLOS ONE

Additional Editor Comments (if provided):

Thank you for your submission. The comments of the reviewers did not suggest additional experiments, but rather changes to the editing and writing of the manuscript. As such, in your revision you should focus on the writing changes requested without concern for additional experiments, especially in light of the COVID-19 pandemic. We look forward to receiving your revised manuscript!

Journal Requirements:

2. In the Methods section of your manuscript, please ensure you have reported:

-the geographic coordinates where sample collection took place;

-what time of year the collection took place;

-whether you had all required permits from local authorities or landowners to perform collection;

-detailed descriptions of all preparation steps (including times and temperatures), especially for the leaf drying and grinding steps;

-how quality assurance was carried out on the extraction process;

-what other ingredients were present in the extract that was administered to the animals; and

-whether (and how) mice were euthanised at the end of the experiment.

Reviewers' comments:

Reviewer's Responses to Questions

**Comments to the Author**

1. Is the manuscript technically sound, and do the data support the conclusions?

Reviewer #1: Partly

Reviewer #2: No

2. Has the statistical analysis been performed appropriately and rigorously? 

Reviewer #1: Yes

Reviewer #2: No

3. Have the authors made all data underlying the findings in their manuscript fully available?

Reviewer #1: Yes

Reviewer #2: Yes

4. Is the manuscript presented in an intelligible fashion and written in standard English?

Reviewer #1: Yes

Reviewer #2: No

5. Review Comments to the Author

Reviewer #1: Summary: The authors present an investigation of one of the active ingredients in the plant, Peperomia pellucida, apparently widely used in the tropics for the treatment of a variety of ailments. Their goal is to identify the mechanism of action of the compound, Pellucidin A, in several models of pain. They demonstrate no increase in locomotor activity of Pellucidin A in the Open Field test, a dose-dependent decrease in number of writhes in the acetic acid model, no effect in the first phase of the formalin model but an effect at high doses in the second (inflammatory) phase. This study provided insights into the antinociceptive application of Pellucidin A.

Major comments:

- It is unclear why the authors believe that the hot plate test is a measure of supraspinal antinociception. Which pain model did they use in this test?

- The results shown in Figure 8 are confusing. If PA has a synergistic effect with L-NAME doesn’t that imply that it does not work through the same mechanism (i.e. does not act on iNOS?). If PA works through COX1 or COX2 inhibition, then the combination with L-NAME may enhance the effect by suppressing two different inflammatory pathways that promote writhing. It is possible that the effect of COX1 or COX2 inhibition is already at its maximum with indomethacin and NS-398 and therefore there is no additional effect of combining with PA. This needs to be re-considered and discussed because the conclusions made are not supported by the data presented.

Minor comments:

- What is an ARC2 compound? Can this be defined up front?

- While CFA causes a biphasic response, I have never seen it called a “biphasic stimulus”. Would suggest referring to this as CFA to avoid confusion.

- Given that readers outside of the tropics may not have ever seen this plant, would recommend that an image/photo is provided.

- Were there 6 mice per group per dose for Pellucidin? Or did each mouse receive all three doses?

- In the Open Field Methods the language suggests that mice received diazepam AND Pellucidin. Please clarify that diazepam was used as a control to decrease locomotor activity in the Methods (this is nicely explained in the Results already).

- Would suggest re-ordering the figures in order to keep all of the acetic acid writhing experiments together (Figure 8 should follow Figure 4)

- Please correct “sintase” in line 162, page 8 to “synthase”. Please add the drugs mentioned in 2.7 to 2.5 with the vendor names of each drug.

- The title of 3.2 should be changed to “Locomotor activity assessment”.

- We suggest combining Figures 5 and 6.

Reviewer #2: This is a very interesting paper that attempts to determine the antinociceptive activity of Pellucidin A, a potential bioactivity compound from Prperomia genus. The authors extracted this compound from the crude extract and characterized using appropriate methodology. Then they performed several behavioral assays to determine the antinociceptive effect of Pellucidin A. They found that treatment with Pellucidin A reduced acidic acid- and formalin (phase II)-induced pain, but very little impact on thermal pain (hot-plate test). By co-treatment with the inhibitors for COX2 and NOS, they demonstrated that the NO pathway might be the potential mechanism for the antinociceptive activity of Pellucidin A. At the end, the authors performed the molecular docking analysis and found that this compound showed good affinity to bind the NOS and COX-2 binding sites.

This study did demonstrate some significance and novelty. However, this manuscript is poorly written and lacks several key details. The data presented in this study prevents solid conclusions from being drawn and the title they are using. Several key information is missing in the methods session.

Major compulsory revisions

1) It would be great to indicate the gender of mice used in this study.

2) How was the Pellucidin A solution prepared? If there was a stock solution made for all the experiments, what was the solvent?

3) Based on the information from 2.5, the indomethacin was dissolved in 5% NaHCO3 and Tween 80 plus 0.9% NaCl. In the Formalin Test (2.8), the drugs/compounds used in this experiment were in different solvents (Pellucidin A was in saline, unknown solvent for morphine, and indomethacin was in a mixed solution).

4) The dosages of Pellucidin A through the whole paper were not consistent (mg/kg in most of the places, % in some of the places).

5) Provide the time points when the behavior tests were conducted for Phase I and Phase II in the formalin assay. Why were mice treated with drugs/compounds before the induction of pain?

6) In the Hot-plate test, 30 seconds is the normal cut-off to avoid tissue damage. However, the data in Fig. 7 indicated that the authors used a much longer time (morphine group) for this assay. Please provide a rationale.

7) For the mechanistic experiments, provide a rationale for why mice were co-treated with the inhibitors for NOS or COX-2 with Pellucidin A (inhibitor + inhibitor). It is necessary to conduct another set of experiments using the activators for NOS or COX-2 with or without Pellucidin A.

8) Please clarify the statistical analysis. One-way ANOVA or two-way ANOVA? It is also very difficult to follow the statistical analysis results in the figures. It would be helpful to indicate the p < 0.05, p < 0.01, and p < 0.001 differently in the figures. In Fig. 7, it is very confusing how the data were analyzed, and what does the “*” mean. In Fig. 8, the data should be compared between the groups with and without Pellucidin A.

9) Are “Pellucidin A” and “Pellucidina A” the same?

10) In general, the whole paper was poorly written. There are many sentences sound non-scientific, such as sentences start in Lines 50, 55, 170, 223, 247,271,280, …. The numbers were presented without units (Lines 270 and 295), which are meaningless. The manuscript has to be edited by a native speaker of English.

Minor essential revisions

There a lot of minor essential revisions through the paper. Spell out abbreviations when they are shown for the first time, such “NO” in Line 58, i.p., and s.c..

6. PLOS authors have the option to publish the peer review history of their article (what does this mean?). If published, this will include your full peer review and any attached files.

Reviewer #1: No

Reviewer #2: No

---

## [Author Response · Author response to Decision Letter 0]

8 May 2020

Reviewer #1: Summary: The authors present an investigation of one of the active ingredients in the plant, Peperomia pellucida, apparently widely used in the tropics for the treatment of a variety of ailments. Their goal is to identify the mechanism of action of the compound, Pellucidin A, in several models of pain. They demonstrate no increase in locomotor activity of Pellucidin A in the Open Field test, a dose-dependent decrease in number of writhes in the acetic acid model, no effect in the first phase of the formalin model but an effect at high doses in the second (inflammatory) phase. This study provided insights into the antinociceptive application of Pellucidin A.

Major comments:

- It is unclear why the authors believe that the hot plate test is a measure of supraspinal antinociception. Which pain model did they use in this test?

Response:

The hot plate test has been widely used as a measure of supraspinal antinoception in several studies (Lamberts, 2013, Pasternak, 2001). The main features that support to use of the hotplate test as a measure of antinocipection are:

First, in the hotplate test there is a clear increase in morphine antinociception which is associated with supraspinal activity. As showing by Lamberts et al, 2013, mutants mice para Gαo subunits, exhibited a naltrexone-sensitive enhancement of baseline latency in both the hot-plate and warm-water tail-withdrawal tests. In the hot-plate test, a measure of supraspinal nociception, morphine antinociception was increased, and this was associated with an increased ability of opioids to inhibit presynaptic GABA neurotransmission in the periaqueductal gray. 

Second, systemic morphine acts at both spinal and supraspinal sites, including the periaqueductal gray (PAG), and so activates a variety of µ -opioid receptor (MORs). Such MORs may represent different receptor variants (Pasternak, 2001), and them morphine is usually the drug used as control as our data. Therefore, based on these factors, we used the hotplate test as a reliable a reproducible test for assessing supraspinal antinocicpetion. 

- The results shown in Figure 8 are confusing. If PA has a synergistic effect with L-NAME doesn’t that imply that it does not work through the same mechanism (i.e. does not act on iNOS?). If PA works through COX1 or COX2 inhibition, then the combination with L-NAME may enhance the effect by suppressing two different inflammatory pathways that promote writhing. It is possible that the effect of COX1 or COX2 inhibition is already at its maximum with indomethacin and NS-398 and therefore there is no additional effect of combining with PA. This needs to be re-considered and discussed because the conclusions made are not supported by the data presented.

Response: 

1) PA+L-NAME act on same mechanism (iNOS) 

The reviewer suggests that a possible synergistic effect of PA and L-NAME would imply a similar mechanism. We agree, our paper is the first paper evaluating the isolated compound of Peperomia pellucida, Pellucidin A. in in silico methods. De Fatima, 2004; Alves NSF, 2019 Bayma JD, 2000 have study the Peperomia pellucida extract, showing the analgesic and anti-inflammatory activities. 

2) PA works via COX1 or COX2, then adding indomethacin or NS 398 there could be enhancement of suppression of these pathways

The reviewer suggests that a possible synergistic effect of PA and COX-1 and COX-2 would imply a similar mechanism. The Figure 8 was our major question, because PA could be presenting a synergistic effect. This data, induced the in-silico experiment with COX-2, iNOS and eNOS. The in-silico, also demonstrate which COX-2, iNOS and eNOS are potential pharmacologicals targets of Pellucidin A. Thereby, we have three action pathways which Pellucidin A may be acting together or no. Our paper is the first paper evaluating the isolated compound of Peperomia pellucida, Pellucidin A. in in silico methods and also, correlating with in vivo methods.

Minor comments:

- What is an ARC2 compound? Can this be defined up front?

Response: According to Bayma (2000) and Ayafor (1987), ArC2 dimers are considered products of a Diels-Alder reaction between two 2,4,5-trimethoxystyrene units, which is also reported in the literature as occurring in this plant. Differently, Ayafor (1987), names ArC2 dimers as bisnorlignans, which contrasts with what was suggested in his work because bisnorlignan is formed by the loss of two C1 units from a lignan (ArC3 dimer). Therefore, in our study, this substance is considered as an ArC2 dimer and not bisnorlignans.

- While CFA causes a biphasic response, I have never seen it called a “biphasic stimulus”. Would suggest referring to this as CFA to avoid confusion.

Response: We corrected the sentence according to the reviewer’s suggestion. The sentence was changed to: In the biphasic response (p.3, line 50).

- Given that readers outside of the tropics may not have ever seen this plant, would recommend that an image/photo is provided.

Response: According to the reviewer’s suggestion, we included a phototgraph to illustrate this plant (p.5, line 104).

- Were there 6 mice per group per dose for Pellucidin? Or did each mouse receive all three doses?

Response: we confirm that there were 6 mice per group andTo improve clarity, we corrected the sentence to (Male Swiss mice, n=6/group) were treated with different doses of Pellucidin A (0.5; 1; or 5 mg/kg, i.p.) (p.8, line 157)

- In the Open Field Methods the language suggests that mice received diazepam AND Pellucidin. Please clarify that diazepam was used as a control to decrease locomotor activity in the Methods (this is nicely explained in the Results already).

Response: We corrected the sentence according to the reviewer’s suggestion. The sentence was inserted: Diazepam was used as a control to decrease locomotor activity (Crawley, 1985) (p.8, line 159 and p. 11, line 245).

- Would suggest re-ordering the figures in order to keep all of the acetic acid writhing experiments together (Figure 8 should follow Figure 4)

Response: We appreciate the reviewer’s suggestions, but we chose to keep figure 8 (Now Figure 7) before the silico figures (Figures 8 and 9), because it provides a framework for understanding the mechanism. 

- Please correct “sintase” in line 162, page 8 to “synthase”. Please add the drugs mentioned in 2.7 to 2.5 with the vendor names of each drug.

Response: We corrected the sentence according to the reviewer’s suggestion. The sentence was changed to: nitric oxide synthase (L-NAME 5 mg/kg) (p.8, line 168). The drugs mentioned in 2.7 to 2.5 with the vendor names of each drug were added (p.7, line 142).

- The title of 3.2 should be changed to “Locomotor activity assessment”.

Response: We corrected the sentence according to the reviewer’s suggestion. The sentence was changed to change to Locomotor activity assessment (p.11, line 240). 

- We suggest combining Figures 5 and 6.

Reply: We agree. We combine the Figure according to the reviewer’s suggestion, now is Figure 5A and B 

Reviewer #2: This is a very interesting paper that attempts to determine the antinociceptive activity of Pellucidin A, a potential bioactivity compound from Prperomia genus. The authors extracted this compound from the crude extract and characterized using appropriate methodology. Then they performed several behavioral assays to determine the antinociceptive effect of Pellucidin A. They found that treatment with Pellucidin A reduced acidic acid- and formalin (phase II)-induced pain, but very little impact on thermal pain (hot-plate test). By co-treatment with the inhibitors for COX2 and NOS, they demonstrated that the NO pathway might be the potential mechanism for the antinociceptive activity of Pellucidin A. At the end, the authors performed the molecular docking analysis and found that this compound showed good affinity to bind the NOS and COX-2 binding sites. This study did demonstrate some significance and novelty. However, this manuscript is poorly written and lacks several key details. The data presented in this study prevents solid conclusions from being drawn and the title they are using. Several key information is missing in the methods session.

Major compulsory revisions

1) It would be great to indicate the gender of mice used in this study.

Response: We corrected the sentence according to the reviewer’s suggestion. The sentence was changed to adult male mice (p.7, line 132)

2) How was the Pellucidin A solution prepared? If there was a stock solution made for all the experiments, what was the solvent?

Response: As requested by the reviewer, we included details on the preparation of the Pellucidin A solution. We added the sentence “Finally, the Pellucidin A solution was prepared before each experiment Tween 80 plus 0.9% NaCl. The final concentration of Tween 80 did not exceed 5% and did not cause any effect per se.” (p.7, line 147)

3) Based on the information from 2.5, the indomethacin was dissolved in 5% NaHCO3 and Tween 80 plus 0.9% NaCl. In the Formalin Test (2.8), the drugs/compounds used in this experiment were in different solvents (Pellucidin A was in saline, unknown solvent for morphine, and indomethacin was in a mixed solution).

Response: We clarified the drug solutions (p.7, line 147)

4) The dosages of Pellucidin A through the whole paper were not consistent (mg/kg in most of the places, % in some of the places).

Response: We agree. We corrected the sentence according to the reviewer’s suggestion.

5) Provide the time points when the behavior tests were conducted for Phase I and Phase II in the formalin assay. Why were mice treated with drugs/compounds before the induction of pain?

Response: Usually, the administration of anti-inflammatory drugs is performed before the phlogistic agent (Formalin solution). This method enables one to distinguish the site of action of analgesics at time point: whether it is central, peripheral or both central and peripheral. Our results show close agreement with the classification of analgesics by the inflamed foot method as described by Sawada, 2014 or Shibata, 1989.

6) In the Hot-plate test, 30 seconds is the normal cut-off to avoid tissue damage. However, the data in Fig. 7 indicated that the authors used a much longer time (morphine group) for this assay. Please provide a rationale.

Response: We agree with the reviewer that the cut-off time for the hot plate test can also be lower than the used in our study as within the range of 20-40s at 48 and 55C as shown in Woolfe and Macdonald, 1944. However, in our study we used a cut-off of 60s as shown in Lamberts et al. 2013. 

The hot plate test, first described in 1944, can be used to determine heat thresholds in mice and rats (Woolfe and Macdonald, 1944). That methodology has two steps:

1- The mice are tested to first cut-off. In this test the animals are submitted a temperature between 48 and 55 ºC, mice usually respond within 20-40s (That is the first cut-off before the test). Thus, all animals that will be tested have the same pain threshold before drug test injection.

2- The mice are tested to drug test cut-off. The methodology used here was the same, used in lamberts 2013, normally the cut-off used is 60s. That time is standard, because Morphine cut-off is 60s, and that time is the point to take out animals from thermal stimuli. Since morphine is the standard drug it is important to compare the results of the new drug with the effects of morphine (See Fig 6)

7) For the mechanistic experiments, provide a rationale for why mice were co-treated with the inhibitors for NOS or COX-2 with Pellucidin A (inhibitor + inhibitor). It is necessary to conduct another set of experiments using the activators for NOS or COX-2 with or without Pellucidin A.

Response: We agree. That moment we could not have NOS or COX-2 agonist or a western blotting setup to quantify these proteins. 

As demonstrated in the figure 7, the data did not show difference between COX- inhibitor and COX-2 inhibitor plus PA. Probably, for the COX-2 system, Pellucidin A shows strong interaction on uncompetitive binding sites, on the same tendency of IMN crystal inhibitor (Figure 9) or COX-inhibitor in vivo in the same binding sites (Figure 7). New experiments must be carried out in the future. We would need to do an enzymatic kinetic study. We are looking to new collaborations to do.

On the other hand, in figure 7, treatment with Pellucidin A showed a synergetic effect with L-NAME treatment, inhibiting the contractions by 96%, when compared to control. Mice treated only with L-NAME presented 48% of contraction inhibition. This result follows our molecular docking model, which showed that Pellucidin A could bind to NOS, showing less energy demand to bind to iNOS (Figure 8 and Table 2). 

8) Please clarify the statistical analysis. One-way ANOVA or two-way ANOVA? It is also very difficult to follow the statistical analysis results in the figures. It would be helpful to indicate the p < 0.05, p < 0.01, and p < 0.001 differently in the figures. In Fig. 7, it is very confusing how the data were analyzed, and what does the “*” mean. In Fig. 8, the data should be compared between the groups with and without Pellucidin A.

Response: We agree. We corrected the sentence according to the reviewer’s suggestion.

9) Are “Pellucidin A” and “Pellucidina A” the same?

Response: We corrected Pellucidina A to Pellucidin A.

10) In general, the whole paper was poorly written. There are many sentences sound non-scientific, such as sentences start in Lines 50, 55, 170, 223, 247,271,280,… The numbers were presented without units (Lines 270 and 295), which are meaningless. The manuscript has to be edited by a native speaker of English.

Response: We have made corrections to improve the readability of the manuscript. Moreover, as suggested by the reviewer, the manuscript has been edited by a native speaker of English. 

Minor essential revisions

There a lot of minor essential revisions through the paper. Spell out abbreviations when they are shown for the first time, such “NO” in Line 58, i.p., and s.c..

Response: We corrected the abbreviations as indicated by the reviewer. (p.8, line 157)

---

## [Decision Letter · Decision Letter 1]

12 Jun 2020

PONE-D-20-04760R1

Pellucidin A promotes antinociceptive activity by peripheral mechanisms inhibiting COX-2 and NOS:  In Vivo, and In Silico Study.

PLOS ONE

Dear Dr. Bastos,

Thank you for submitting your manuscript to PLOS ONE. After careful consideration, we feel that it has merit but does not fully meet PLOS ONE’s publication criteria as it currently stands. Therefore, we invite you to submit a revised version of the manuscript that addresses the points raised during the review process.

We look forward to receiving your revised manuscript.

Kind regards,

John M. Streicher, Ph.D.

Academic Editor

PLOS ONE

Additional Editor Comments (if provided):

Thank you for your resubmission. If you can quickly address the minor revisions requested by Reviewer 1, I will accept the manuscript without further rounds of review.

Reviewers' comments:

Reviewer's Responses to Questions

**Comments to the Author**

1. If the authors have adequately addressed your comments raised in a previous round of review and you feel that this manuscript is now acceptable for publication, you may indicate that here to bypass the “Comments to the Author” section, enter your conflict of interest statement in the “Confidential to Editor” section, and submit your "Accept" recommendation.

Reviewer #1: (No Response)

Reviewer #2: All comments have been addressed

2. Is the manuscript technically sound, and do the data support the conclusions?

Reviewer #1: Yes

Reviewer #2: Yes

3. Has the statistical analysis been performed appropriately and rigorously? 

Reviewer #1: Yes

Reviewer #2: Yes

4. Have the authors made all data underlying the findings in their manuscript fully available?

Reviewer #1: Yes

Reviewer #2: Yes

5. Is the manuscript presented in an intelligible fashion and written in standard English?

Reviewer #1: Yes

Reviewer #2: Yes

6. Review Comments to the Author

Reviewer #1: While the authors have addressed some of the comments, there remain a few important corrections that need to be made:

1. I still don't see any mention of which pain model was used in the mice in the hot plate test. Are these mice uninjured? formalin-treated? please specify.

2. Page 14, line 304. it still states "insert p value here"

3. The Discussion is extremely long and confusing. It would benefit from reorganization for clarity. For example, iNOS is mentioned in multiple different paragraphs that are not linked. Please go through and make sure there is a logical flow to the Discussion in order to highlight only the most salient points of the paper.

Reviewer #2: The authors have addressed all the concerns raised in the original submission. There is no further comments.

7. PLOS authors have the option to publish the peer review history of their article (what does this mean?). If published, this will include your full peer review and any attached files.

Reviewer #1: No

Reviewer #2: No

---

## [Author Response · Author response to Decision Letter 1]

24 Jul 2020

Response Letter

Reviewer #1: While the authors have addressed some of the comments, there remain a few important corrections that need to be made:

1. I still don't see any mention of which pain model was used in the mice in the hot plate test. Are these mice uninjured? formalin-treated? please specify.

Response:

 The hot plate test has been widely used as a measure of supraspinal antinoception in several studies (Lamberts, 2013, Pasternak, 2001). The thermal or chemical agents just induce pain without injure mice . The time between the placement of the mouse on the platform and shaking, licking of the paws or jumping was recorded as the hot-plate latency, to clarify in Hot-plate test (2.9), Materials and Methods, line 200, you can read “Licking or jumping is a rapid response to painful thermal stimuli that is a direct indicator of nociceptive threshold”, page 09. Also, was explain in formalin test. 

2. Page 14, line 304. it still states "insert p value here"

Response:

We corrected the sentence according to the reviewer’s suggestion. The sentence was changed and values were insert 

3. The Discussion is extremely long and confusing. It would benefit from reorganization for clarity. For example, iNOS is mentioned in multiple different paragraphs that are not linked. Please go through and make sure there is a logical flow to the Discussion in order to highlight only the most salient points of the paper.

Response:

We corrected the discussion according to the reviewer’s suggestion.

Reviewer #2: The authors have addressed all the concerns raised in the original submission. There is no further comments.

Response:

 Thanks

---

## [Editor Report · Decision Letter 2]

26 Aug 2020

Pellucidin A promotes antinociceptive activity by peripheral mechanisms inhibiting COX-2 and NOS:  In Vivo, and In Silico Study.

PONE-D-20-04760R2

Dear Dr. Bastos,

We’re pleased to inform you that your manuscript has been judged scientifically suitable for publication and will be formally accepted for publication once it meets all outstanding technical requirements.

Kind regards,

John M. Streicher, Ph.D.

Academic Editor

PLOS ONE
---

## [Editor Report · Acceptance letter]

31 Aug 2020

PONE-D-20-04760R2 

Pellucidin A promotes antinociceptive activity by peripheral mechanisms inhibiting COX-2 and NOS: In Vivo and In Silico Study. 

Dear Dr. Bastos:

I'm pleased to inform you that your manuscript has been deemed suitable for publication in PLOS ONE. Congratulations! Your manuscript is now with our production department. 

Kind regards, 

on behalf of

Dr. John M. Streicher 

Academic Editor

PLOS ONE